**Data Availability Statement:** There are restrictions to data sharing but these are based on the various legislative restrictions for data containing

# Predictors for one-year outcomes of cardiorespiratory fitness and cardiovascular risk factor control after cardiac rehabilitation in elderly patients: The EU-CaRE study

**Prisca Eser**[1], **Thimo Marcin**[1,2], **Eva Prescott**[3], **Leonie F. Prins**[4], **Evelien Kolkman**[4], **Wendy Bruins**[5], **Astrid E. van der Velde**[5], **Carlos Peña Gil**[6], **Marie-Christine Iliou**[7], **Diego Ardissino**[8], **Uwe Zeymer**[9], **Esther P. Meindersma**[10], **Arnoud W. J. Van'tHof**[5,11,12], **Ed P. de Kluiver**[5], **Matthias Wilhelm**[1] *

**1** Department of Cardiology, Inselspital, Bern University Hospital, University of Bern, Bern, Switzerland, **2** Graduate School for Health Sciences, University of Bern, Bern, Switzerland, **3** Department of Cardiology, Bispebjerg Frederiksberg University Hospital, Copenhagen, Denmark, **4** Diagram B.V., Zwolle, The Netherlands, **5** Isala Heart Centre, Zwolle, The Netherlands, **6** Department of Cardiology, Hospital Clínico Universitario de Santiago, SERGAS, FIDIS, CIBER CV, University of Santiago de Compostela, A Coruña, Spain, **7** Department of Cardiac Rehabilitation, Assistance Publique Hopitaux de Paris, Paris, France, **8** Department of Cardiology, Parma University Hospital, Parma, Italy, **9** Klinikum Ludwigshafen and Institut für Herzinfarktforschung Ludwigshafen, Ludwigshafen, Germany, **10** Department of Cardiology, Radboud University, Nijmegen, The Netherlands, **11** Department of Cardiology, Maastricht University Medical Center, Maastricht, The Netherlands, **12** Department of Cardiology, Zuyderland Medical Center, Heerlen, The Netherlands

* Matthias.wilhelm@insel.ch

## Abstract

### Introduction

Studies on effectiveness of cardiac rehabilitation (CR) in elderly cardiovascular disease patients are rare, and it is unknown, which patients benefit most. We aimed to identify predictors for 1-year outcomes of cardiorespiratory fitness and CV risk factor (CVRF) control in patients after completing CR programs offered across seven European countries.

### Methods

Cardiovascular disease patients with minimal age 65 years who participated in comprehensive CR were included in this observational study. Peak oxygen uptake ($VO_2$), body mass index (BMI), resting systolic blood pressure (BPsys), and low-density lipoprotein-cholesterol (LDL-C) were assessed before CR (T0), at termination of CR (T1), and 12 months after start of CR (T2). Predictors for changes were identified by multivariate regression models.

### Results

Data was available from 1241 out of 1633 EU-CaRE patients. The strongest predictor for improvement in peak $VO_2$ was open chest surgery, with a nearly four-fold increase in surgery compared to non-surgery patients. In patients after surgery, age, female sex, physical inactivity and time from index event to T0 were negative predictors for improvement in peak

potentially identifying or sensitive patient information of the involved eight countries and the lack of informed consent to international data sharing. Requests for de-identified aggregated data may be sent to info@diagram-zwolle.nl, Diagram B. V., contract research organization, Dokter Stolteweg 96, 8025 AZ Zwolle, The Netherlands, on behalf of Prof. Arnoud Van't Hoff, Department of Cardiology, Maastricht University Medical Center, and Cardiovascular Research Institute Maastricht (CARIM), the Netherlands.

**Funding:** For the Swiss consortium partner (TM, PE, MW), funding was received by the Swiss State Secretariat for Education, Research and Innovation under contract number 15.0139. All other authors received funding by the European Union's Horizon 2020 research and innovation programme under grant agreement No 634439. The funders had no role in study design, data collection and analysis, decision to publish, or preparation of the manuscript. None of the commercial partners who provided financial support to any of the coauthors had any role in the perception, conduction, analysis, interpretation nor dissemination of this study.

**Competing interests:** AWJVH reports grants from Medtronic, grants and personal fees from Astra Zeneca, outside the submitted work, UZ reports grants and personal fees from Astra Zeneca, grants and personal fees from Bayer, personal fees from Boehringer Ingelheim, grants and personal fees from BMS, personal fees from Daiichi Sankyo, personal fees from Eli Lilly, grants and personal fees from Novartis, grants and personal fees from MSD, personal fees from Trommsdorf, personal fees from Amgen, outside the submitted work. LP and EK work for Diagram B.V., a contract research organization. These commercial affiliations do not alter our adherence to all PLOS ONE policies on sharing data and materials. All other authors have no Conflict of Interest to declare.

VO$_2$. In patients without surgery, previous acute coronary syndrome and higher exercise capacity at T0 were the only negative predictors. Neither number of attended training sessions nor duration of CR were significantly associated with change in peak VO$_2$. Non-surgery patients were more likely to achieve risk factor targets (BPsys, LDL-C, BMI) than surgery patients.

## Conclusions

In a previously understudied population of elderly CR patients, time between index event and start of CR in surgery and disease severity in non-surgery patients were the most important predictors for long-term improvement of peak VO$_2$. Non-surgery patients had better CVRF control.

## Background

Comprehensive cardiac rehabilitation programmes (CR) have the potential to improve cardio-respiratory fitness (CRF) [1], cardiovascular risk factors (CVRF) [2], and reduce hospitalisations and cardiovascular mortality [2–6]. However, long-term CVRF control in patients with acute and chronic coronary syndromes remains poor across Europe [7]. Studies on short- and long-term effectiveness of CR generally suffer from underrepresentation of elderly patients, particularly patients older than 80 years of age.

The mean age of classical CR studies was around 60 to 62 years, and predictors for a lesser improvement in CRF have been found to be older age, female sex, increased body mass index (BMI), smoking and presence of diabetes mellitus (DM) [8–10]. Whether the same factors are responsible for effectiveness in elderly CR patients, has not been investigated so far.

EU-CaRE was a European project focusing on the effectiveness and sustainability of CR programs in the elderly (65 years or above). The main objective of EU-CaRE was to obtain the evidence base to improve, tailor and optimise CR programmes regarding sustainable effectiveness, cost-effectiveness and participation level. EU-CaRE involved eight participating CR sites in seven countries (Denmark, France, Germany, Italy, the Netherlands, Spain and Switzerland) [11].

The purpose of the present study was to identify predictors for CR effectiveness in terms of improvement in CRF and CVRF control, namely goal-directed therapy of low-density-lipoprotein-cholesterol (LDL-C), blood pressure (BP), weight, and HbA1c in patients with DM. Depending on the nature of identified predictors, better results by CR may be achieved through appropriate adaptation of the CR programmes.

## Methods

The EU-CaRE observational study compared different CR programmes provided to elderly cardiac patients at eight European sites in Bern, Copenhagen, Ludwigshafen, Paris, Parma, Nijmegen, Santiago de Compostela and Zwolle. We aimed at including a total of 1760 patients equivalent of 220 patients from each site. The CR program offered at each site has been described previously [12].

### Study population

The study population and baseline data have been reported previously [11]. Briefly, elderly (65 +) patients participating in CR of one of the eight European centres after coronary artery

bypass grafting (CABG), percutaneous coronary intervention (PCI) or without revascularization as well as after percutaneous or surgical valve replacement (HVR) were included. Patients were assessed at baseline before commencing CR (T0), after completing the CR program (T1) and at 1-year follow-up (T2).

The study was approved by the lead ethics committee (Medisch Ethische Toetsingscommissie at Isala, Netherlands) and all relevant medical ethics committees, registered at trialregister.nl (NTR5306). The participants gave written informed consent before they were included in the study.

## Data collection

Recorded information included demographics, index event, socioeconomic factors, medical history including co-morbidity, and clinical information such as weight, blood pressure (BP), resting heart rate and CRF (cardiopulmonary exercise testing, CPET), medication, and patient reported outcomes which included physical activity in terms of number of days per week with at least moderate physical activity of minimally 30 min. Obesity was defined as body mass index (BMI) $\geq$30 kg/m$^2$. Exercise capacity was assessed by CPET. Number of attended training sessions were monitored at each site, offered sessions ranged from 10 to 36 between sites. Lag time of CR start was defined as days between revascularisation or valve treatment and CR entry visit (T0). Date of first diagnosis was used to calculate lag time in stable CAD patients without surgical or percutaneous intervention. CR duration was defined as days between CR start (T0) to CR conclusion (T1). Details on the data collected have been provided elsewhere [11–14].

CPETs were performed on cycle ergometers with spirometry, electrocardiogramme registration and blood pressure measurement using local certified equipment and a common protocol according to current recommendations [15]. A warm-up at 5 Watt was performed for 3 min. The ramp protocol was individualized to achieve the optimal test duration of 8–12 min until exhaustion. We aimed to surpass the anaerobic threshold and a respiratory quotient (RQ) of >1.1. CPET data was analysed in the CPET core lab in Bern. Peak oxygen uptake (VO$_2$) was determined from raw data files using MATLAB software from MathWorks®. Gas measures were excluded from the analysis in case of suspected mask leakage or equipment failure or if the ramp duration was less than 3 min. In these cases, VO$_2$ was calculated with a formula using the maximum Watt [16].

## Outcomes

Predictors were sought for the primary outcome, changes in CRF (T2-T0), namely change in peak VO$_2$ [ml/kg/min]. Predictors were also sought for secondary outcomes, namely for CVRF control at T1 and T2, including the achievement of target levels according to current guidelines [15] as follows: systolic BP (BPsys) < 140 mmHg, LDL-C < 1.8 mmol/l, BMI < 30 kg/m$^2$ (non-obesity) or lowering body weight by $\geq$ 5%, and HbA1c in diabetic patients < 53 mmol/mol.

## Statistical analysis

All statistics were performed with R (Version 3.5.1, R Core Team, 2017).

Robust linear mixed models were performed for change in peak VO$_2$ [ml/min/kg] between T0 and T2 (function rlmer from package robustlmm version 2.3). The following factors were included as predictor variables: Age, sex and peak exercise capacity at T0, comorbidities, cardiovascular risk factors and characteristics of CR, namely, lag time of CR and CR duration and self-reported physical activity level during follow-up. Centre was entered into the model as a

random factor. Further, factors from a list of cardiovascular risk factors and comorbidities (provided in S1 Table) were selected by backward step-wise regression. Non-significant variables were eliminated from the model manually by individually removing the variable with the largest p-value until any insignificant (p>0.01) parameters were removed from the model. This was done to avoid removing non-significant variables which had a large effect on other significant variables. To adjust at least for some of the multiple testing, alpha was arbitrarily set at 0.01 for all analyses. The resulting model was also performed with an imputed data set. Missing values were imputed with multiple imputation (5 imputations, 10 iterations) with predictive mean matching for all variables using the function mice from mice package (v3.4.0).

Logistic regression models were performed using the glmer function of the lme4 package (v1.1–21) for CVRF control (non-/achievement of target levels) at T1 and T2. CR centres were included as random factors because we assumed that counselling and education on cardiovascular risk factor treatment may have been important for these variables and were likely to have differed between centres. The same predictor variables as used for the linear models were included as fixed factors, except that peak CRF at T0 and reported physical activity at T2 were omitted because they were not considered relevant, instead, relevant medications were included.

## Results

A total of 1633 patients were included (T0), 1523 (93%) completed the end of CR assessment (T1), 1457 (89%) the one-year follow-up (T2) and 1429 (88%) attended all three examinations, as reported previously [13,14,17]. The study flow is shown in Fig 1. For the present study, data of maximally 1457 patients were included due to missing data in any of the parameters included into the models. Baseline (pre-CR) characteristics of the study population are shown in Table 1 split into patients who had an open chest surgery preceding CR and those who had no intervention or minimally invasive interventions. A detailed description of the primary and secondary outcomes has been presented previously [17].

### Predictors for primary outcome

Since patients with open chest surgery (CABG or surgical HVR) had lower CRF in the first few days and weeks following surgery but steeper recovery thereafter compared to patients with only minimally invasive or no procedures (PCI, stable angina or percutaneous HVR patients) [13,14], we performed the models of the primary outcome for these two populations separately. Due to some missing data, 779 non-surgery (including 23 percutaneous HVR) and 469 surgery (including 100 HVR) patients were included. However, this reduced data set was comparable to the total data set with regard to change in peak $VO_2$ and age.

Mean increase in peak $VO_2$ from T0 to T2 was 4.4 (SD 3.7) ml/kg/min in surgery and 1.4 (SD 2.3) ml/kg/min in non-surgery patients, whereby values were lower in surgery patients before CR and comparable to non-surgery patients after one year (Table 1). Results from the robust mixed linear models for the two split populations (surgery and non-surgery patients) are presented in Fig 2. The following factors were found to significantly contribute to a lesser change in peak $VO_2$ between T0 and T2 in the surgery patients: Older age, female sex, weight gain, physical inactivity at T2 and lag time to CR start. In non-surgery patients, significant factors were previous ACS (before the index event of this study), weight change, and a lower starting exercise capacity (Fig 2). The largest difference in change in peak $VO_2$ was found between the surgery and non-surgery groups, with the surgery patients improving 3.06 (99% CI 2.57–3.55) ml/kg/min more than the non-surgery patients. In the surgery patients, males improved 2.07 (99% CI 1.04–3.10) ml/kg/min more than females. A decade in age reduced the

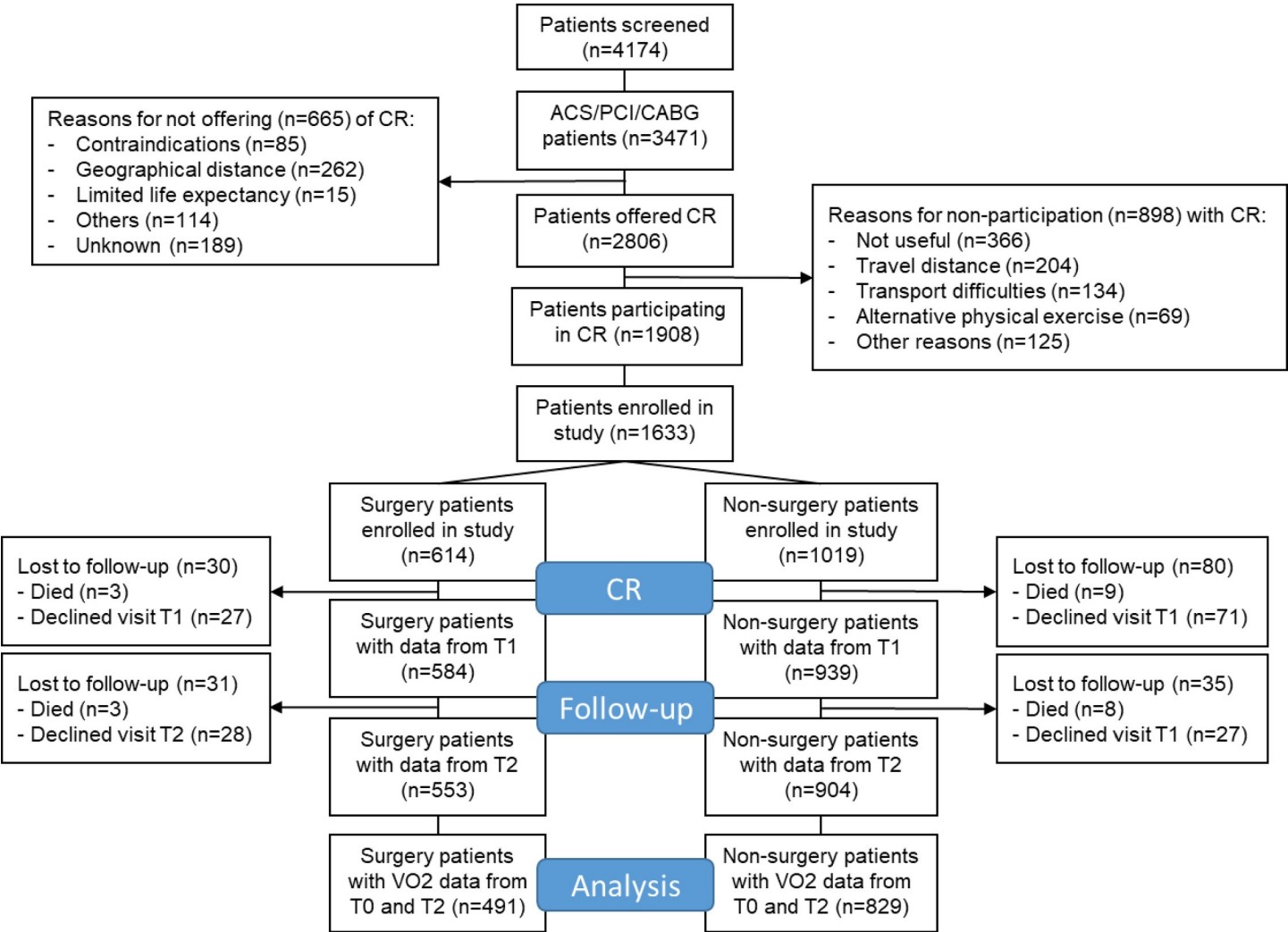

**Fig 1. Study flow.** Analysis refers to primary outcome change in peak VO$_2$ from baseline to 1-year follow-up.

improvement in peak VO$_2$ by approximately 1.12 (99% CI -1.86 to -0.04) ml/kg/min, a lag time to start of CR by an additional 10 days reduced the improvement in peak VO$_2$ by approximately 0.30 (99% CI -0.55 to -0.04) ml/kg/min, and an additional day per week with 30 min of at least moderate physical activity at T2 increased the change in peak VO$_2$ by 0.25 (99% CI 0.09–0.41) ml/kg/min in the surgery patients. Amongst the non-surgery patients, a previous ACS decreased the change in peak VO$_2$ by 0.73 (99% CI -1.37 to -0.10) ml/kg/min, and a higher exercise capacity at T0 by 1 Watt/kg reduced the change in peak VO$_2$ by 0.67 (99% CI -1.28 to -0.05) ml/kg/min. A body weight increase by 1 kg reduced the change in peak VO$_2$ relative to body weight by approximately -0.15 (99% CI -0.23 to -0.06) and -0.18 (99% CI -0.25 to -0.12) ml/kg/min in surgical and non-surgical patients, respectively. Neither, duration of CR nor number of attended training sessions had a significant effect in either population. The same models with the imputed data resulted in very similar effect sizes (S1 Fig).

## Secondary endpoints

Deviances of all models were acceptable and all smaller than 1.5 times the degrees of freedom.

**Table 1. Baseline characteristics of surgery and non-surgery patients.**

| Parameter | Surgery (614) | Non-surgery (1019) |
|---|---|---|
| *Personal characteristics* | | |
| Female sex (%) | 126 (20.5%) | 248 (24.3%) |
| Age[yrs] | 72.8±5.1 | 72.9±5.7 |
| BMI [kg/m$^2$] | 26.7±4.0 | 27.5±4.2 |
| BP systolic [mmHg] | 122.2±17.7 | 126.6±17.0 |
| BP diastolic [mmHg] | 72.5±11.0 | 72.9±9.9 |
| LDL-C [mol/l] | 2.17±0.76 | 2.04±0.70 |
| HbA1c [mmol/mol] | 40.2±8.6 | 42.0±9.0 |
| *Index intervention* | | |
| VHD | 133 (21.7%) | 33 (3.2%) |
| CABG | 481 (78.3%) | 0 (0%) |
| PCI | 0 (0%) | 890 (87.3%) |
| Stable angina | 0 (0%) | 96 (9.4%) |
| *Cardiovascular risk factors* | | |
| Smoking | 49 (8.0%) | 105 (10.3%) |
| Days with >30 min physical activity | 3.9±2.8 | 3.7±2.8 |
| Previous ACS | 81 (13.2%) | 223 (22.0%) |
| Hypertension | 428 (69.8%) | 678 (66.9%) |
| Hypercholesteremia | 419 (68.4%) | 770 (66.7%) |
| Family history of CVD | 239 (39.2%) | 255 (25.1%) |
| Diabetes mellitus | 146 (23.8%) | 253 (24.8%) |
| *Comorbidities* | | |
| Nephropathy | 43 (7.0%) | 81 (8.0%) |
| Chronic heart failure | 22 (3.6%) | 20 (2.0%) |
| Peripheral arterial disease | 53 (8.6%) | 73 (7.2%) |
| Atrial fibrillation | 49 (8.0%) | 65 (6.4%) |
| *Medication* | | |
| Beta blocker | 524 (85.3%) | 803 (78.8%) |
| Statins | 500 (81.4%) | 960 (94.2%) |
| ACE/ARB | 330 (53.7%) | 774 (76.0%) |
| *CR characteristics* | | |
| Time lag index event-CR start [days] | 21 (12, 46)* | 32 (18,52)* |
| Duration of CR [days] | 24 (17, 54*) | 86 (27, 134)* |
| Number of attended training sessions | 14.2±6.2 | 16.9±9.3 |
| Peak VO$_2$ at CR start | 14.4±4.3 | 16.9±4.9 |
| Peak VO$_2$ after 1 year | 19.1±5.4 | 18.6±5.4 |

*Data presented as median (Q1, Q3) due to non-parametric distribution.

ACE/ARB, angiotensin converting enzyme inhibitors/angiotensin II receptor blockers; ACS, acute coronary syndrome; BMI, body mass index; BP, blood pressure; CABG, coronary artery bypass grafting; CR, cardiac rehabilitation, CVD, cardiovascular disease; diabetes mellitus; LDL-C, low-density lipoprotein cholesterol; HbA1c, glycated haemoglobin; PCI, percutaneous coronary intervention; VHD, valvular heart disease; VO$_2$, oxygen uptake.

**BPsys.** BPsys remained stable overall between T0 and T1, with 78.9% of patients reaching target BPsys at T0 and 83.6% at T1. Age was the only predictor for a higher proportion of patients not meeting target BPsys at T1, with a decade reducing the chance of meeting target BPsys by 43%. At T2, only 73.7% reached target BPsys, at T2 BPsys was higher by a

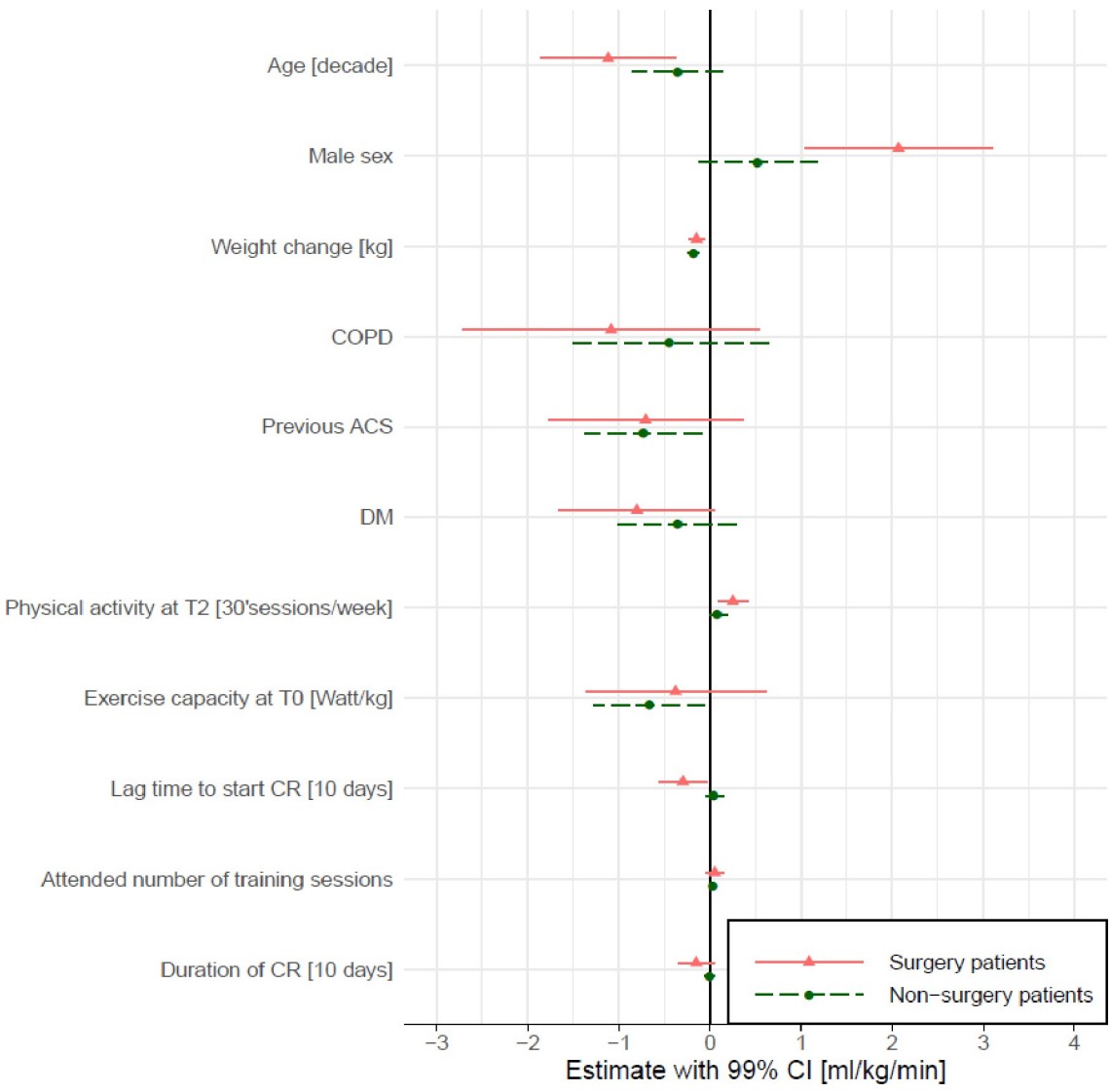

**Fig 2. Predictors for change in peak VO$_2$ in the 468 surgery and 773 non-surgery patients.** Shown are estimates and 99% confidence intervals of included parameters from the separate robust linear mixed regression models for change in peak VO$_2$ from T0 to T2 for the two sub-populations. CRF at T0, attended number of training sessions, inactivity, lag time to CR start and duration of CR were included a priori. The other variables were significant variables in the model for the pooled data but shown here for the surgery and non-surgery populations separately. Centre was included as random factor. COPD, chronic obstructive pulmonary disease; ACS, acute coronary syndrome; DM, diabetes mellitus; CR, cardiac rehabilitation.

median of 5 mmHg (Q3, Q4: -6, 15 mmHg) compared to T1. Negative predictors for meeting target levels in BPsys at T2 were older age and index open chest surgery, with surgery patients having an odds ratio of 0.7 for achieving target BPsys compared to non-surgery patients (Fig 3A).

**Weight.** BMI remained stable overall between T0 and T1, but increased by a median BMI of 0.29 kg/m2 (Q3, Q4: -0.33, 0.93 kg/m2) at T2. 23.2% of all patients were obese at T0, while 20.9% were obese at T1 and 23.6% at T2. Eleven patients were obese at T1 but had decreased their weight from T0 by at least 5%. At T2, there were no obese patients who had decreased their weight by at least 5% between T1 and T2. As expected, BMI at T0 was strongly predictive

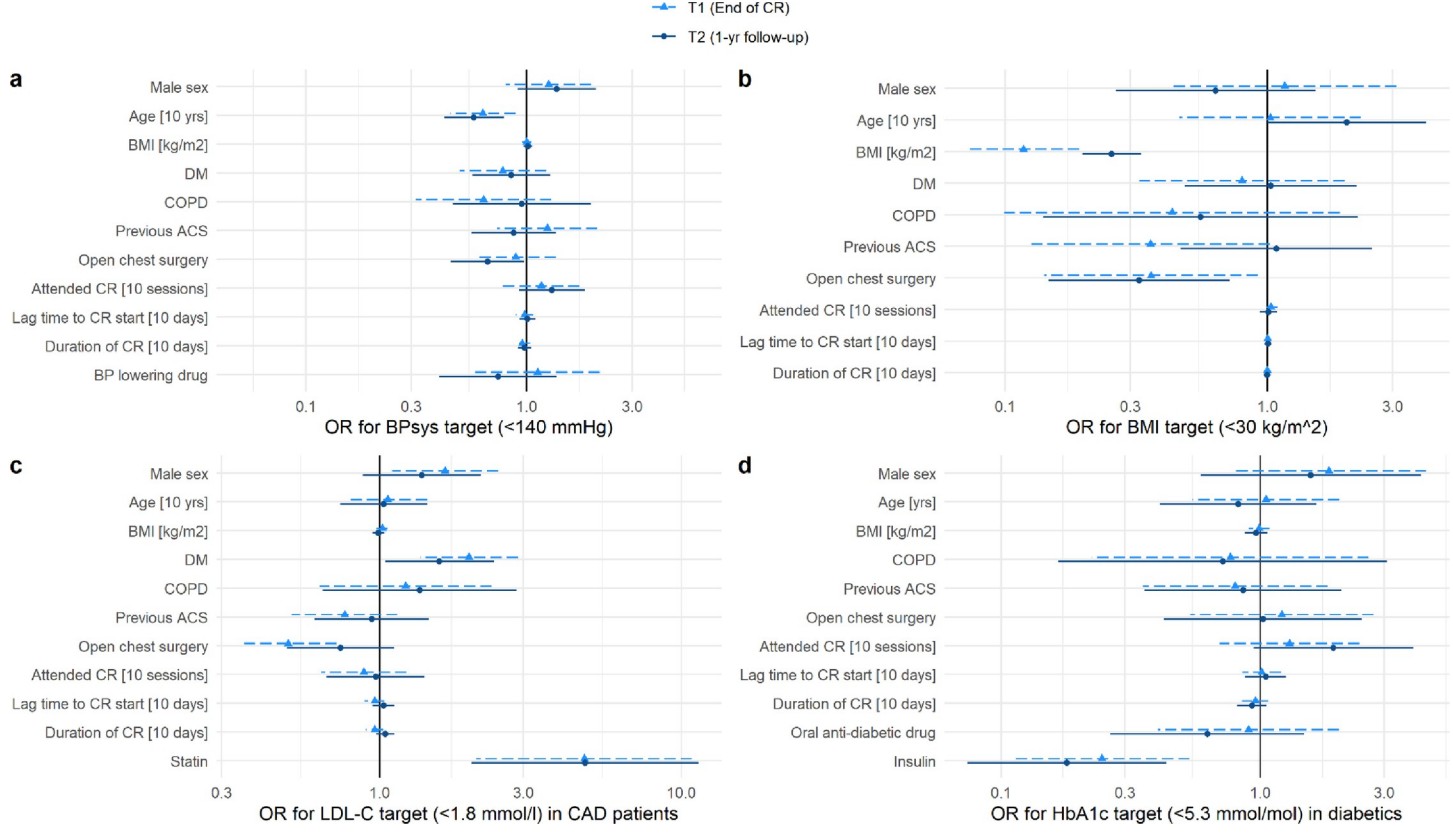

**Fig 3. Predictors for reaching target levels of secondary outcome parameters at T1 and T2.** Shown are odds ratios and 99% confidence intervals of included parameters from the logistic mixed regression model for reaching target BPsys (a), target BMI (b), target LDL-C in CAD patients (c), and target HbA1c in diabetic patients (d). Centres (random factor) are not shown. Number of patients included in the models for T1 and T2 were 1451 and 1305 for BPsys, 1455 and 1311 for BMI, 1203 and 1058 for LDL-C (excluding VHD patients), and 314 and 280 for HbA1c in diabetic patients. BMI, body mass index; DM, diabetes mellitus; COPD, chronic obstructive pulmonary disease; ACS, acute coronary syndrome; CR, cardiac rehabilitation.

of BMI at T1 and T2. Surgery patients had an odds ratio of approximately 0.3 to reach target levels of BMI compared to non-surgery patients (Fig 3B).

**LCL-C.** Amongst the CAD patients (excluding patients with VHD), LDL-C declined from T0 to T1 by a median of 0.09 mmol/l (Q3, Q4: -0.37, 0.20 mmol/l), with 39.5% reaching target levels at T0 and 43.4% at T1. Between T1 and T2 there was an increase in LDL-C by a median of 0.08 mmol/l (Q3, Q4: -0.20, 0.42 mmol/l), with only 35.4% of all patients reaching target levels at T2. At T1, surgery patients had an odds ratio of 0.5 to reach target levels compared to non-surgery patients, and males had an odds ratio of 1.3 compared to females (Fig 3C). Patients with DM were 2.0 and 1.6 more likely to reach target LDL-C at T1 and T2, respectively. Patients taking statins had an odds ratio of 4.8 to reach target LDL-C at T1 and T2 compared to those not taking statins (7.1%).

**HbA1c in patients with diabetes.** HbA1c in patients with DM decreased between T0 and T1 by 1.0 mmol/mol (Q3, Q4: -4.0, 3.0 mmol/mol) with 62.2% and 66.6% reaching target levels of <53 mmol/mol at T0 and T1, respectively, and increased from T1 to T2 by a median 1.1 mmol/mol (Q3, Q4: -2.19, 5.47 mmol/mol) with 59.1% reaching target levels at T2. The only significant predictor for patients reaching target levels was intake of insulin at both time points, with an odds ratio of 0.24 for reaching target HbA1c in patients taking insulin at T1 and 0.18 at T2 (Fig 3D).

## Discussion

The main predictor for sustainable effectiveness of the primary outcome peak $VO_2$ was index intervention, with patients with open chest surgery starting at lower peak $VO_2$ values than those with no or only percutaneous procedures but reaching comparable values at 1-year follow-up [13]. In non-surgery patients, advanced disease with previous ACS, weight increase, and a higher exercise capacity at baseline were the only predictors for a lesser improvement in peak $VO_2$, while in surgery patients, female sex, older age, greater lag time, physical inactivity and weight increase were negative predictors. Non-surgery patients had better outcomes with regard to BPsys and LDL-C than surgery patients. Male CAD patients and patients with DM had better outcomes with regard to LDL-C.

### Primary outcome

We observed only small changes in peak $VO_2$ in non-surgery patients of 1/3 metabolic equivalent (MET), with weight change, previous ACS, and exercise capacity at T0 being significant predictors. The effect of weight change was due to the relative unit of CRF in $VO_2$ per body mass. Previous ACS can be taken as synonymous with more advanced disease progression. The inverse relationship between exercise capacity at baseline and increase in peak $VO_2$ during CR may be partly due to patients with lower fitness having a greater potential to improve, and partly due to a phenomenon termed "regression towards the mean" [18].

In surgery patients, on the other hand, changes in peak $VO_2$ between T0 and T2 were just over 1 MET due to lower peak $VO_2$ at T0 but comparable values at T2 compared to non-surgery patients. In our patients, mean improvement in peak $VO_2$ of surgery patients was almost 4-fold the improvement of non-surgery patients. As mentioned in two previous studies, the reason for this may be lower levels of CRF of surgery patients due to bedrest and surgery-related restrictions in ventilation [19,20]. In fact, a randomized training study in CABG patients found the same improvement in peak $VO_2$ in the exercise and control groups, indicating that recovery takes place also in the absence of a structured exercise-based CR programme [21]. Significant predictors for worse outcome in the surgery patients were older age, female sex, inactivity at T2 and lag time to start of CR. An additional decade in age was related to a reduced recovery in peak $VO_2$ by 1.1 ml/kg/min, while in non-surgery patients the reduction was only 0.4 ml/kg/min, suggesting that older age may not only reduce the effect of a physical training stimulus but also have an adverse effect on the healing process from surgery [22]. A previous study on patients undergoing CR has found older age to be a negative predictor for improvement in CRF [9], and another study found a smaller increase in CRF in elderly compared to younger patients [23]. However, other studies have found similar or even greater improvements in peak $VO_2$ with increasing age [24]. The level of physical activity at 1-year follow-up was positively associated with increase in peak $VO_2$ in our surgery patients. While our data is inconclusive with regard to the direction of a potentially causal relationship, CR has been found to increase physical activity compared to control treatments [25].

Lag time to CR start was inversely related to change in peak $VO_2$ whereby the relationship was exponential rather than linear [13]. Surgery patients who started CR within the first 30 days after surgery had a mean increase in peak $VO_2$ of 5.5±3.9 ml/kg/min (congruent with a lower starting peak $VO_2$), while those starting after 30 days only had a mean increase of 2.9 ±3.2 ml/kg/min, and from 60 days onward it stabilized at 1.7±2.7 ml/kg/min, which was similar to the improvement of non-surgery patients. It is unlikely that patients benefited less from CR if they started CR late after surgery, rather, a late start with CR did not coincide with the steepest recovery process early after surgery. When controlled for lag time, duration of the exercise programme, on the other hand, was not associated with change in peak $VO_2$, this is

remarkable given the fact that prescribed programmes ranged from 2 to 36 weeks. A similar conclusion was drawn by a recent meta-regression analysis, which found exercise intervention intensity to be the only predictive factor for the difference in CRF between randomized exercise-based CR and control groups [26]. Training intensity was monitored in a subset of patients and its influence on change in peak $VO_2$ was reported elsewhere [27].

A surprising result of the present study was the much smaller increase in peak $VO_2$ in the 75 female surgery patients compared to the 393 surgery male patients after the multivariate adjustment. Recovery in physical function [28] and wound healing has been found impeded in female CABG patients compared to males [29]. We cannot exclude that indications for CABG and/or surgical valve replacement were different for male and female patients.

The effect of DM has been addressed in a separate publication [13].

## Secondary outcomes

Target values for BPsys were less likely to be reached by surgery patients and the odds for reaching target were approximately 30% lower for each additional decade higher age.

Regarding BMI, again surgery patients had a smaller odds ratio for reaching BMI target at both time points. Additionally, a decade of higher age was related to a two-fold higher odds ratio of reaching these BMI targets. With 24% obesity, the proportion of obese patients was considerably lower in our elderly population compared to the documented 38% obese patients in the EUROASPIRE population with a mean age of 63 years [7].

Male CAD patients and CAD patients with DM were more likely to achieve LDL-C target values, possibly because dyslipidemia was treated more aggressively in these latter patients. The previously documented acute drop in LDL-C after acute myocardial infarction [30] or cardiac surgery [31] with subsequent recovery may explain the odds ratio of 0.5 for surgery patients to reach LDL-C target levels at T1. A small percentage (7.3%) of our CAD patients was not on statins.

## Strengths and limitations

The main strength of the study is the inclusion of a large and heterogenous elderly patient population of seven European countries, reflecting daily practice of these centres. A further strength is the available data on the heterogenous lag times to start of the CR programme, which in elderly surgery patients seems to be the main determinant for improvement in CRF.

One of the main limitations is that lag time after event to start of CR and CR duration were not independent of centres, therefore the predictive effects of these time variables may be under- or overestimated due to additional specific CR programme or cultural/ethnic effects.

In elderly people, it is difficult to achieve maximal CPETs due to many physical ailments such as orthopedic disabilities. However, when the model for primary outcome was adjusted for change in respiratory exchange ratio (a parameter commonly used for level of achieved exhaustion), effects remained comparable.

The patients lost to follow-up (T2) had with 13.8 (95% CI 13.3–14.3) ml/kg/min a significantly lower baseline peak $VO_2$ than patients completing follow-up with 16.4 (95% CI 16.3–16.6) ml/kg/min. Therefore, our results may be more applicable to fitter patients.

## Conclusion

The present study revealed specific predictors for clinically relevant 1-year outcomes in an elderly CR population, an understudied patient group. It demonstrated a large difference in the recovery process between patients with and without open chest surgery. Older age and female sex were associated with poorer recovery in peak $VO_2$ in surgery patients. Neither

length of the CR program nor the number of CR training sessions were associated with long-term improvement of CRF while levels of physical activity at T2 were relevant. CVRF control was better in non-surgery patients. This highlights the importance of individualized physical activity counselling and CVRF management in comprehensive CR [32].

## Supporting information

**S1 Checklist. TREND checklist EU-CaRE.**
(PDF)

**S1 Fig. Predictors for change in peak VO2 for 614 surgery and 1019 non-surgery patients.** Shown are pooled estimates and 99% confidence intervals (using rubins rule) from the robust linear mixed models over the multiple imputed data (5 imputations with predictive mean matching). CRF at T0, attended number of training sessions, inactivity, lag time to CR start and duration of CR were included a priori. The other variables were significant variables in the model for the pooled data but shown here for the surgery and non-surgery populations separately. Centre was included as random factor. COPD, chronic obstructive pulmonary disease; ACS, acute coronary syndrome; DM, diabetes mellitus; CR, cardiac rehabilitation.
(DOCX)

**S1 Table. List of variables entered into robust linear models.**
(DOCX)

## Acknowledgments

We greatly appreciated the statistical advice by Prof. J. Hüsler, Institute of Mathematical Statistics and Actuarial Science of the University of Berne.

## Author Contributions

**Conceptualization:** Prisca Eser, Eva Prescott, Astrid E. van der Velde, Carlos Peña Gil, Marie-Christine Iliou, Diego Ardissino, Uwe Zeymer, Arnoud W. J. Van'tHof, Ed P. de Kluiver, Matthias Wilhelm.

**Data curation:** Prisca Eser, Thimo Marcin, Eva Prescott, Leonie F. Prins, Evelien Kolkman, Carlos Peña Gil.

**Formal analysis:** Prisca Eser, Thimo Marcin.

**Funding acquisition:** Eva Prescott, Wendy Bruins, Astrid E. van der Velde, Arnoud W. J. Van'tHof, Ed P. de Kluiver.

**Investigation:** Prisca Eser, Thimo Marcin, Eva Prescott, Leonie F. Prins, Wendy Bruins, Astrid E. van der Velde, Carlos Peña Gil, Marie-Christine Iliou, Diego Ardissino, Uwe Zeymer, Esther P. Meindersma, Arnoud W. J. Van'tHof, Ed P. de Kluiver, Matthias Wilhelm.

**Methodology:** Prisca Eser, Thimo Marcin, Eva Prescott, Leonie F. Prins, Astrid E. van der Velde, Marie-Christine Iliou, Diego Ardissino, Uwe Zeymer, Esther P. Meindersma, Arnoud W. J. Van'tHof, Ed P. de Kluiver, Matthias Wilhelm.

**Project administration:** Thimo Marcin, Eva Prescott, Leonie F. Prins, Evelien Kolkman, Wendy Bruins, Astrid E. van der Velde, Carlos Peña Gil, Marie-Christine Iliou, Diego Ardissino, Uwe Zeymer, Esther P. Meindersma, Arnoud W. J. Van'tHof, Ed P. de Kluiver, Matthias Wilhelm.

**Supervision:** Arnoud W. J. Van'tHof, Ed P. de Kluiver, Matthias Wilhelm.

**Validation:** Thimo Marcin, Leonie F. Prins, Evelien Kolkman.

**Visualization:** Thimo Marcin.

**Writing – original draft:** Prisca Eser.

**Writing – review & editing:** Prisca Eser, Thimo Marcin, Eva Prescott, Leonie F. Prins, Evelien Kolkman, Wendy Bruins, Astrid E. van der Velde, Carlos Peña Gil, Marie-Christine Iliou, Diego Ardissino, Uwe Zeymer, Esther P. Meindersma, Arnoud W. J. Van'tHof, Ed P. de Kluiver, Matthias Wilhelm.

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
