## [Decision Letter · Decision Letter 0]

14 Apr 2021

PONE-D-20-33363

Predictors for one-year outcomes of cardiorespiratory fitness and cardiovascular risk factor control after cardiac rehabilitation in elderly patients: the EU-CaRE study

PLOS ONE

Dear Dr. ESER,

Thank you for submitting your manuscript to PLOS ONE. After careful consideration, we feel that it has merit but does not fully meet PLOS ONE’s publication criteria as it currently stands. Therefore, we invite you to submit a revised version of the manuscript that addresses the points raised during the review process.

Please read and respond to all of the peer review comments. You should provide a point-by-point response to explain any changes you have (or have not) made to the original article and be as specific as possible in your responses.

We look forward to receiving your revised manuscript.

Kind regards,

Gerson Cipriano Jr., PT, MsC, Ph.D.

Academic Editor

PLOS ONE

Journal Requirements:

"The study was approved by all relevant medical ethics committees of all participating centres, and registered at trialregister.nl (NTR5306".   

'AWJvH reports grants from Medtronic, grants and personal fees from Astra Zeneca, outside the submitted work, UZ reports grants and personal fees from Astra Zeneca, grants and personal fees from Bayer, personal fees from Boehringer Ingelheim, grants and personal fees from BMS, personal fees from Daiichi Sankyo, personal fees from Eli Lilly, grants and personal fees from Novartis, grants and personal fees from MSD, personal fees from Trommsdorf, personal fees from Amgen, outside the submitted work.'

a. Please confirm that this does not alter your adherence to all PLOS ONE policies on sharing data and materials, by including the following statement: "This does not alter our adherence to  PLOS ONE policies on sharing data and materials.” (as detailed online in our guide for authors http://journals.plos.org/plosone/s/competing-interests).  If there are restrictions on sharing of data and/or materials, please state these.

Please note that we cannot proceed with consideration of your article until this information has been declared.

5. Please amend the manuscript submission data (via Edit Submission) to include author P. de Kluiver.

6. Please include captions for your Supporting Information files at the end of your manuscript, and update any in-text citations to match accordingly. Please see our Supporting Information guidelines for more information: http://journals.plos.org/plosone/s/supporting-information

Reviewers' comments:

Reviewer's Responses to Questions

**Comments to the Author**

1. Is the manuscript technically sound, and do the data support the conclusions?

Reviewer #1: Yes

Reviewer #2: Yes

Reviewer #3: Yes

2. Has the statistical analysis been performed appropriately and rigorously? 

Reviewer #1: Yes

Reviewer #2: Yes

Reviewer #3: Yes

3. Have the authors made all data underlying the findings in their manuscript fully available?

Reviewer #1: Yes

Reviewer #2: Yes

Reviewer #3: No

4. Is the manuscript presented in an intelligible fashion and written in standard English?

Reviewer #1: Yes

Reviewer #2: Yes

Reviewer #3: Yes

5. Review Comments to the Author

Reviewer #1: Interesting paper.

Abstract; at firsr glance it remains unclear how patients were selected. Which were inclusion criteria?

methods: definition of elderly should be added

methods/results>a relevant difference between patients after CABG/cardiac surgery and after PCI is present

methods: it should be added how variavles were selected

methods: primary end point and secondary end points should be added

methods: sample size calculation should be added

Reviewer #2: This is a powerful paper which contributes to expand the knowledge about cardiovascular rehabilitation in elderly patients. In this paper, the authors analyzed the effect of cardiac rehabilitation in the enhancement of traditional cardiovascular risk factors – LDL-cholesterol, blood pressure, weight, and HbA1c in patients with DM - and improvement of functional capacity (peak oxygen consumption – peak VO2).

(Also see attached document with comments)

Reviewer #3: I commend the authors on this work and after having reviewed the manuscript have the following queries:

General comments:

Please verify first expansion of all abbreviations and ensure that is done only once

Specific comments:

1. Did the sites of recruitment have an influence on your results? This could be a confounder

2. I understand the numbers of those not completing CR are small - but did you attempt to identify any factors within that group as well (if they have CPET values)?

3. Were any functional test performed in the elderly? This could be used as surrogates among those in whom CPET was not possible

6. PLOS authors have the option to publish the peer review history of their article (what does this mean?). If published, this will include your full peer review and any attached files.

Reviewer #1: **Yes: **Fabrizio D'Ascenzo

Reviewer #2: **Yes: **Alexandra Correa Gervazoni Balbuena de Lima

Reviewer #3: No

---

## [Author Response · Author response to Decision Letter 0]

5 May 2021

Point-by-point responses to reviewers’ comments

We thank the reviewers for their constructive comments and hope we have addressed them accordingly. Please find below our responses to the reviewers’ comments in italic font.

Review Comments to the Author

Reviewer #1: Interesting paper.

Abstract; at firsr glance it remains unclear how patients were selected. Which were inclusion criteria?

Detailed inclusion criteria have been published previously in: Prescott E, Meindersma EP, van der Velde AE, Gonzalez-Juanatey JR, Iliou MC, Ardissino D, Zoccai GB, Zeymer U, Prins LF, Van't Hof AW, Wilhelm M, de Kluiver EP. A EUropean study on effectiveness and sustainability of current Cardiac Rehabilitation programmes in the Elderly: Design of the EU-CaRE randomised controlled trial. Eur J Prev Cardiol. 2016 Oct;23(2 suppl):27-40. And

Prescott E, Mikkelsen N, Holdgaard A, Eser P, Marcin T, Wilhelm M, et al. Cardiac rehabilitation in the elderly patient in eight rehabilitation units in Western Europe: Baseline data from the EU-CaRE multicentre observational study. European journal of preventive cardiology. 2019;26(10):1052-63.

Additionally, we have amended the following sentence on line 97, page 6, as follows: “Briefly, elderly (65+) patients participating in CR of one of the eight European centres after coronary artery bypass grafting (CABG), percutaneous coronary intervention (PCI) or without revascularization as well as after percutaneous or surgical valve replacement (HVR) were included.”

methods: definition of elderly should be added

Elderly was defined as patients aged ≥65 years (please see sentence cited above).

methods/results>a relevant difference between patients after CABG/cardiac surgery and after PCI is present

methods: it should be added how variavles were selected

Selection of variables is described on page 8, line 138: “Robust linear mixed models were performed for change in peak VO2 [ml/min/kg] between T0 and T2 (function rlmer from package robustlmm version 2.3). The following factors were included as predictor variables: Age, sex and peak exercise capacity at T0, comorbidities, cardiovascular risk factors and characteristics of CR, namely, lag time of CR and CR duration and self-reported physical activity level during follow-up. Centre was entered into the model as a random factor. Further factors from a list of cardiovascular risk factors and comorbidities (provided in S 1 Table) by backward selection. Non-significant variables were eliminated from the model manually by individually removing the variable with the largest p-value until any insignificant (p>0.01) parameters were removed from the model. This was done to avoid removing non-significant variables which had a large effect on other significant variables.”

methods: primary end point and secondary end points should be added

Primary and secondary end points were the predictors for the following primary and secondary outcomes (page 7, line 130): “Predictors were sought for the primary outcome, changes in CRF (T2-T0), namely change in peak VO2 [ml/kg/min]. Predictors were also sought for secondary outcomes, namely for CVRF control at T1 and T2, including the achievement of target levels according to current guidelines [15] as follows: BPsys < 140 mmHg, LDL-C < 1.8 mmol/l, BMI < 30 kg/m2 (non-obesity) or lowering body weight by ≥5%, and HbA1c in diabetic patients < 53 mmol/mol.”

methods: sample size calculation should be added

The sample size calculation for the EU-CaRE observational study is stated on page 6, line 93. “We aimed at including a total of 1760 patients equivalent of 220 patients from each site..” 

Reviewer #2: This is a powerful paper which contributes to expand the knowledge about cardiovascular rehabilitation in elderly patients. In this paper, the authors analyzed the effect of cardiac rehabilitation in the enhancement of traditional cardiovascular risk factors – LDL-cholesterol, blood pressure, weight, and HbA1c in patients with DM - and improvement of functional capacity (peak oxygen consumption – peak VO2).

(Also see attached document with comments)

Reviewer #3: I commend the authors on this work and after having reviewed the manuscript have the following queries:

General comments:

Please verify first expansion of all abbreviations and ensure that is done only once

Thank you for pointing this out, we have corrected some missing or duplicated abbreviations.

Specific comments:

1. Did the sites of recruitment have an influence on your results? This could be a confounder

This is a good comment, yes, sites were a confounder, because CR programmes differed in duration and intensity from each other. Further, patient inclusion time into the programmes differed as well, with some centres starting patients in their programmes within days after interventions (stationary programmes) and several weeks or months (ambulatory programmes). However, we wanted to find predictors with regard to patient characteristics, not centre characteristics, which is why we included centres as random factors. The influence of site has been addressed in the publication of the main study results: Prescott E, Eser P, Mikkelsen N, Holdgaard A, Marcin T, Wilhelm M, Gil CP, González-Juanatey JR, Moatemri F, Iliou MC, Schneider S, Schromm E, Zeymer U, Meindersma EP, Crocamo A, Ardissino D, Kolkman EK, Prins LF, van der Velde AE, Van't Hof AW, de Kluiver EP. Cardiac rehabilitation of elderly patients in eight rehabilitation units in western Europe: Outcome data from the EU-CaRE multi-centre observational study. Eur J Prev Cardiol. 2020 Nov;27(16):1716-1729. 

2. I understand the numbers of those not completing CR are small - but did you attempt to identify any factors within that group as well (if they have CPET values)?

Unfortunately, none of the centres had CPET data from patients not undergoing CR.

3. Were any functional test performed in the elderly? This could be used as surrogates among those in whom CPET was not possible

Indeed, 6-minute-walking tests were performed in patients who were too weak to perform CPETs. We have presented detailed results in:

“A CPET was performed by 1534 patients, 48 performed a 6MWT and 51 patients did not have a baseline functional test available.” We have presented these results in detail in:

Prescott E, Eser P, Mikkelsen N, Holdgaard A, Marcin T, Wilhelm M, Gil CP, González-Juanatey JR, Moatemri F, Iliou MC, Schneider S, Schromm E, Zeymer U, Meindersma EP, Crocamo A, Ardissino D, Kolkman EK, Prins LF, van der Velde AE, Van't Hof AW, de Kluiver EP. Cardiac rehabilitation of elderly patients in eight rehabilitation units in western Europe: Outcome data from the EU-CaRE multi-centre observational study. Eur J Prev Cardiol. 2020 Nov;27(16):1716-1729.

---

## [Editor Report · Decision Letter 1]

1 Jun 2021

PONE-D-20-33363R1

Predictors for one-year outcomes of cardiorespiratory fitness and cardiovascular risk factor control after cardiac rehabilitation in elderly patients: the EU-CaRE study

PLOS ONE

Dear Dr. ESER,

Thank you for submitting your manuscript to PLOS ONE. After careful consideration, we feel that it has merit but does not fully meet PLOS ONE’s publication criteria as it currently stands. Therefore, we invite you to submit a revised version of the manuscript that addresses the points raised during the review process.

Although the authors have adequately responded to all reviewers' comments, a complete proofreading review is required.

We look forward to receiving your revised manuscript.

Kind regards,

Gerson Cipriano Jr., PT, MsC, Ph.D.

Academic Editor

PLOS ONE
---

## [Author Response · Author response to Decision Letter 1]

1 Jun 2021

Responses to reviewer 2

Thank you very much for your supportive comments. Please find below our responses to your comments and questions. We hope that we have addressed them according to your expectations.

Questions

Line 92. “We aimed at including a total of 1760 patients equivalent of 220 patients from each site”. 

How was the sample size calculated? In the design paper of the study - A EUropean study on effectiveness and sustainability of current Cardiac Rehabilitation programmes in the Elderly: Design of the EU-CaRE randomised controlled trial. European journal of preventive cardiology. 2016;23(2 suppl):27-40 – it is described the sample size of the Eu-Care RCT (randomised controlled trial) based on the improvement of peak O2 consumption, but not from the observational study.

The EU-CaRE study consisted of an observational study and an RCT. For the observational study, sample size was based on number of expected eldery patients seen each year and consisted of 220 patients per centre. In the methods section on p. 6 it reads: We aimed at including a total of 1760 patients equivalent of 220 patients from each site. The CR program offered at each site has been described previously [12].

Ref. 12 (see above) refers to the description of the CR program only, not to the sample size calculation.

Suggestion of citation

Kim KH, Jang YC, Song MK, Park HK, Choi IS, Han JY in the recent paper “Changes in Aerobic Capacity Over Time in Elderly Patients with Acute Myocardial Infarction During Cardiac Rehabilitation. Ann Rehabil Med. 2020 Feb;44(1):77-84. doi: 10.5535/arm.2020.44.1.77. Epub 2020 Feb 29. PMID: 32130841; PMCID: PMC7056327” found a similar no improvement in functional capacity in elderly patients in CR.

Thank you for pointing out this reference, we have included it in the Discussion section.

Suggestion of limitation

 Almost 50% of CR participants are older adults (>65 years), many of whom are frail or deconditioned. We need to start looking for frailty in elderly patients entering CR programmes and become more familiar with the tools to recognize and evaluate the severity of this condition. Furthermore, we need to better understand whether exercise-based cardiac rehabilitation may change the course and the prognosis of frailty in cardiovascular patients. The benefits of CR are well-suited to counteract the deficits of frailty such as sarcopenia, inactivity, fatigue, cognitive decline, and depression.

We couldn’t agree more!

---

## [Editor Report · Decision Letter 2]

19 Jul 2021

Predictors for one-year outcomes of cardiorespiratory fitness and cardiovascular risk factor control after cardiac rehabilitation in elderly patients: the EU-CaRE study

PONE-D-20-33363R2

Dear Dr. ESER,

We’re pleased to inform you that your manuscript has been judged scientifically suitable for publication and will be formally accepted for publication once it meets all outstanding technical requirements.

Kind regards,

Gerson Cipriano Jr., PT, MsC, Ph.D.

Academic Editor

PLOS ONE
---

## [Editor Report · Acceptance letter]

28 Jul 2021

PONE-D-20-33363R2 

Predictors for one-year outcomes of cardiorespiratory fitness and cardiovascular risk factor control after cardiac rehabilitation in elderly patients: the EU-CaRE study 

Dear Dr. Wilhelm:

I'm pleased to inform you that your manuscript has been deemed suitable for publication in PLOS ONE. Congratulations! Your manuscript is now with our production department. 

Kind regards, 

on behalf of

Professor Gerson Cipriano Jr. 

Academic Editor

PLOS ONE